# Characters Are Like Faces

**Haoyu Deng**[1], **Zhaoteng Ye**[1], **Yule Duan**[1*]
[1]University of Electronic Science and Technology of China
{haoyu_deng, 2020010912015 , duanyll}@std.uestc.edu.cn

## Abstract

There are over 100,000 characters in Chinese, though only four thousand of them are used in our daily life. However for cultural researchers, they interact with those Rarely Used Characters (RUCs) frequently. It would facilitate using these RUCs for them with Optical Character Recognition (OCR) technology. Nevertheless, the current OCR methods, no matter regression based or classification based, are difficult to recognize such a huge amount of characters. In this work, we simply treat characters like human faces and adopt the MobileFaceNetV3 to recognize over 74,000 Chinese characters included in Unicode. A demo can be seen at http://risingentropy.top/OCR.html. All source code:https://github.com/RisingEntropy/Characters-Are-Like-Faces

## 1 Introduction

The proposal of convolution neuron network (CNN) makes huge affections to both OCR (Long et al. (2020)) and face recognition (Du et al. (2022)) tasks. One thing the two tasks have in common is that they both identify a certain region of an image. However, the output property is distinct. The output of OCR tasks is limited in a static set since the symbols in a language is definite. Face recognition tasks are completely contradict because they require to recognize a person whose face is never learnt by the network and dynamically add or delete faces. For languages like English, only a small amount of characters are used so regression or classification methods work out fine, whereas for languages like Chinese, a huge set of symbols is used that regression or classification methods might not work. In this work, we try using face recognition techniques to address this issue. We utilize the MobileFaceNetV3 (Chen et al. (2018) ), a tiny and light network as the name suggests, which outputs a 64 dimension feature vector. To obtain a better clustered output, we apply the arcface loss (Deng et al. (2022)) to train the network.

## 2 Method

### 2.1 Workflow

The overall workflow is demonstrated in Figure 1. We utilize the MobileFaceNetV3 (Chen et al. (2018)) as our backbone network. The network outputs a 64 dimension vector that represents the feature of the input image. To instruct the network properly extract feature, we adopt the arcface loss (Deng et al. (2022)) which can better guide clustering. The arcface loss handles the feature vector and transforms it into a onehot vector to train the network. Like a standard face recognition application, we first establish a feature vector database using images created from various fonts. The feature vector of a character is the average value of the output of different fonts to better locate the center in the 64D space. To detect a character, the process is simple. Just to obtain the feature vector using the trained MobileFaceNetV3 and search the database to find one record with the minimal $\ell_2$ norm.

### 2.2 Data Generation

We generate our data by rendering images of Chinese characters using different fonts. A challenge of this approach is that no single font covers all the Chinese characters encoded in Unicode. To overcome this limitation and improve generalization performance, we collect 30 of the most widely-used Chinese fonts and select 500 characters as our training set.

---

*Corresponding author

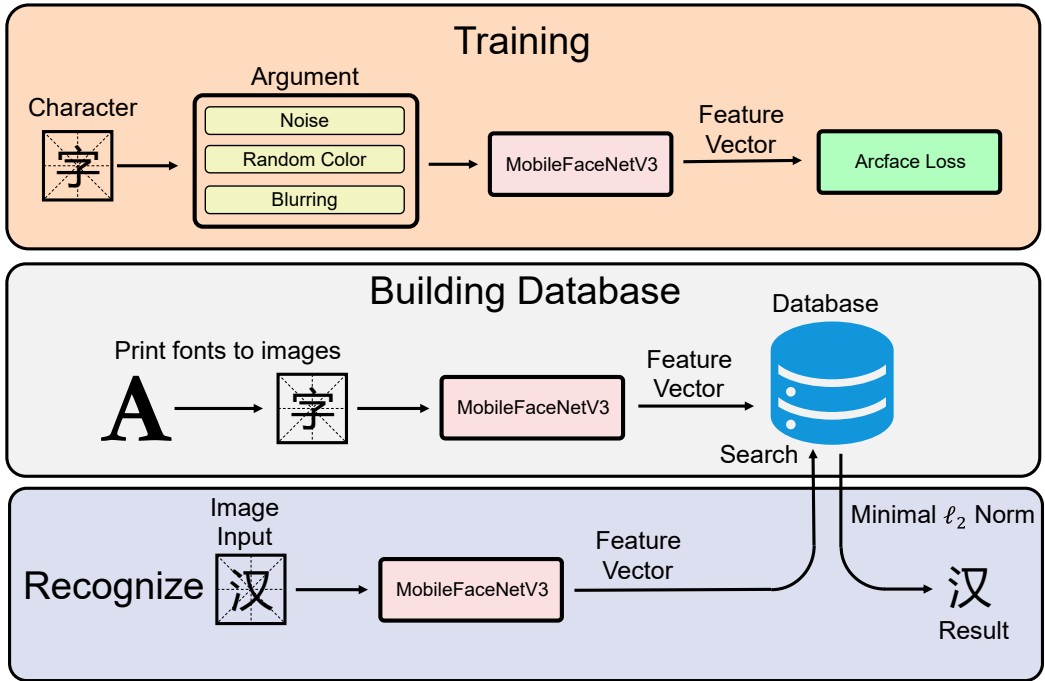

Figure 1: The workflow.

We apply various data augmentation techniques, such as adding noise, random characters, background colors. Additionally, we blur the images by downsampling and then upsampling them to reduce their resolution. However, this may have a negative effect on some complex characters that become indistinguishable from other similar ones after blurring.

## 3 EXPERIMENT

We evaluate this recognition method on various fonts. The experiment result is listed in Table 1. The recognition accuracy is higher than 90%, demonstrating its effectiveness and robustness. What we need to be aware of is that our current model cannot handle handwriting characters since all our data is computer generated. Considering the extraordinary performance of MobileFaceNetV3, we believe that this model can deal with hand writings after training on corresponding datasets. Due to that KaiXinSong-2.1 contains more RUCs than others, for most characters there is only one sample. Therefore, when doing evaluation, the feature vector is nearly the same with the one from database thus increasing the accuracy.

| Font Name | Sample | Test chacacters | Accuracy |
|---|---|---|---|
| KaiXinSong-2.1 | 汉字 | 5000 | 99.98% |
| KaiXinSongA | 汉字 | 5000 | 97.62% |
| HeiTi | 汉字 | 5000 | 97.60% |
| SimSong | 汉字 | 5000 | 99.38% |
| STKAITI | 汉字 | 5000 | 91.82% |
| XiaoBiaoSong | 汉字 | 5000 | 97.62% |

KaiXinSong-2.1 and KaiXinSongA are actually the same in graphs. KaiXinSong-2.1 contains more RUCs but commonly used ones are excluded. KaiXinSongA contains most common used characters and very limited RUCs.

Table 1: Experiment in various fonts.

## 4 CONCLUSION

In this paper, we try using MobileFaceNetV3 to recognize Chinses characters. The experiment results show that the accuracy exceeds 90% on various datasets. We establish a website make it convenient for cultural researchers to recognize RUCs. A shortcoming of our model is that it cannot deal with hand-writing characters. To address this problem, hand-writing character dataset is needed, which is now lacking. Despite this problem, our current model can still free cultural researchers from looking up dictionary when facing printed characters.

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
