# OpenReview forum: "Characters Are Like Faces"
_ICLR.cc/2023/TinyPapers — Submitted to Tiny Papers @ ICLR 2023_

### Official Review · Reviewer_7hpr · 2023-03-28

**Confidence:** 4

**Summary Of Contributions:**

The study proposes using MobileFaceNetV3 for Chinese character recognition, achieving over 90% accuracy on various font datasets.

**Rating:**

Clear, Correct, and Reproducible (CCR): a submission which meets the reviewing criteria

**Strengths And Weaknesses:**

## Strengths:

1. The paper uses  MobileFaceNetV3, a tiny and light network that outputs a 64-dimension feature vector, which is effective in recognizing Chinese characters. It is a perfect use case for transfer learning.
2. The establishment of a website that makes it convenient for cultural researchers to validate Chinese characters.

##  Weakness:

The limitations of the research include the lack of a hand-writing character dataset, which limits the ability of the model to recognize hand-written characters. Additionally, while the recognition accuracy exceeded 90% on various datasets, the model may have difficulty with more complex characters that become indistinguishable from other similar ones after blurring. Also, the use of computer-generated data may not fully represent the variability of real-world handwriting, which could affect the generalizability of the model.



**Suggested Changes:**

1. Including real-world handwriting in the training to solve it for real handwriting would be a great next step.
2. Please consider evaluating the study along with the SSD-Mobilenet implementation.

---

### Official Review · Reviewer_uqpD · 2023-03-29

**Confidence:** 2

**Summary Of Contributions:**

The paper is focused on OCR for Chinese characters, with a focus on rarely used characters. Given the large and diverse set of characters in the Chinese language, the authors propose using a face detection model for the OCR task, which gives promising results.

**Rating:**

Clear, Correct, and Reproducible (CCR): a submission which meets the reviewing criteria

**Strengths And Weaknesses:**

Strengths:

-	The paper is well written with a clearly stated goal and promising results.

-	The model was explained with sufficient detail for this format.

Weaknesses:

-	No baseline model is included. While the accuracy is seemingly respectable, it is not clear from the paper how a traditional OCR approach would have fared. Given that the central claim of the paper is that a face detection model is a better fit than an OCR model, this is necessary.

-	I do not fully understand was comprises the train and test set. The abstract mentions 74000 characters, but the training set is only comprised of 5000. Were these randomly sampled? Is there any differentiation between rare or commonly used characters? Table 1 indicates 5000 characters in the test set, are these different than the training set?


**Suggested Changes:**

-	Provide a baseline model as a point of comparison.

-	Clearly indicate what comprises the training and test sets.

-	Given the focus of the paper is rarely used characters, a brief discussion that focuses on results for those characters specifically would be a worthwhile inclusion.

-	Further proofreading would be ideal. While the paper is readable in its current state, the English grammar is poor in several parts.

---

### Meta-Review · Area_Chair_f6fD · 2023-04-08

**Recommendation:** Invite to present
**Confidence:** 4

**Metareview:**

Based on the two reviews, the paper proposes using a face detection model and MobileFaceNetV3 for OCR of Chinese characters. Review 1 acknowledges the well-written paper, clearly stated goal, and promising results. However, it points out the lack of a baseline model and unclear details about the train and test set. Review 2 also recognizes the effectiveness of MobileFaceNetV3 in recognizing Chinese characters, the establishment of a website for cultural researchers, and the limitations of the research, such as the lack of a handwriting character dataset and potential difficulty with complex characters.

Pros:
- Well-written paper with a clearly stated goal.
- Promising results with the use of face detection and MobileFaceNetV3 for OCR of Chinese characters.
- Establishment of a website for cultural researchers to validate Chinese characters.

Cons:
- Lack of a baseline model to compare the results.
- Unclear details about the train and test set.
- Limitations of the research, such as the lack of a handwriting character dataset and potential difficulty with complex characters.


**Summary:**

The paper's main message is to propose the use of face detection and MobileFaceNetV3 for OCR of Chinese characters, which shows promising results. However, there are limitations to the research, and further evaluation is necessary.

**Reason For Not Giving A Higher Recommendation:**

There are some suggested changes by the reviewers that can be incorporated in the submission. Please refer to the suggested changes sections.


**Reason For Not Giving A Lower Recommendation:**

Based on the reviewers’ comments and recommendation, and my review of the paper, I am leaning towards the positive side.

---

### Meta-Review · Area_Chair_zzwG · 2023-04-08

**Recommendation:** Invite to archive
**Confidence:** 3

**Metareview:**

The paper is interesting and potentially valuable in some ways, but are missing some important details according to both reviewers. The strengths and weaknesses are given below

Strengths:
- Novel approach to OCR
- Dataset creation

Weaknesses:
- Lack of relevant research citation, as mentioned (SSD Mobilenet)
- Lack of baseline model
- Unclear experimentation details (train/test split)

At the time of this meta-review, the website is inaccessible

**Summary:**

The paper proposes the use of a face detection system to detect various Chinese characters and it performs impressively well on a constructed dataset. According to the reviewers, the main idea of the paper is clear and the dataset building is a good addition, but some experimental details are missing

**Comments And Feedback To The Authors:**

1. If there is no exisiting Chinese characters dataset, please consider making the dataset available for research
2. Please include a link to the code and data so the claims can be reproduced independently
3. Include a baseline model to show the improvement by using this system.

**Reason For Not Giving A Higher Recommendation:**

## Clarity

Are the findings communicated clearly and effectively? - **Yes**

Does the paper include appropriate discussion of other relevant literature? - **No**

## Correctness

Are the claims and conclusions justified by the findings? - **Yes**

## Reproducibility

Does the paper describe its methods in such detail that a reader could independently reproduce the findings? - **No**

I do not think the paper is CCR and would benefit from some revisions.



**Reason For Not Giving A Lower Recommendation:**

N/A

---

> ### Author Response · Authors · 2023-04-12
> **Response from the author**
>
> Thanks for your kindly reminding us, the website is fixed now and is accessible. All the code used in training/testing (including the source code of the website)  is released on github: https://github.com/RisingEntropy/Characters-Are-Like-Faces. It's really hard to be too detailed within 2 pages, if you are interested in our work, you may refer to the source code. Thanks again for your reminding us.

---

### Decision · Program_Chairs · 2023-04-10

Invite to archive